# The New GPI-Anchored Protein, SwgA, Is Involved in Nitrogen Metabolism in the Pathogenic Filamentous Fungus *Aspergillus fumigatus*

**DOI:** 10.3390/jof9020256

**Published:** 2023-02-15

**Authors:** Marketa Samalova, Patricia Flamant, Rémi Beau, Mike Bromley, Maryse Moya-Nilges, Thierry Fontaine, Jean-Paul Latgé, Isabelle Mouyna

**Affiliations:** 1Unité des Aspergillus, Département de Mycologie Institut Pasteur, 25-28 rue du Docteur Roux, CEDEX 15, 75724 Paris, France; 2Manchester Fungal Infection Group, Division of Infection, Immunity and Respiratory Medicine, Faculty of Biology, Medicine and Health, University of Manchester, CTF Building, Grafton Street, Manchester M13 9NT, UK; 3Unité Technologie et Service Bioimagerie Ultrastructurale (UTechS UBI), Institut Pasteur, 28 rue du Docteur Roux, 75015 Paris, France

**Keywords:** *Aspergillus fumigatus*, conidia, cell wall, GPI-anchored protein, morphogenesis, nitrogen metabolism

## Abstract

GPI-anchored proteins display very diverse biological (biochemical and immunological) functions. An in silico analysis has revealed that the genome of *Aspergillus fumigatus* contains 86 genes coding for putative GPI-anchored proteins (GPI-APs). Past research has demonstrated the involvement of GPI-APs in cell wall remodeling, virulence, and adhesion. We analyzed a new GPI-anchored protein called SwgA. We showed that this protein is mainly present in the *Clavati* of *Aspergillus* and is absent from yeasts and other molds. The protein, localized in the membrane of *A. fumigatus*, is involved in germination, growth, and morphogenesis, and is associated with nitrogen metabolism and thermosensitivity. *swgA* is controlled by the nitrogen regulator AreA. This current study indicates that GPI-APs have more general functions in fungal metabolism than cell wall biosynthesis.

## 1. Introduction

The post-translational modification of proteins by the addition of a glycosylphosphatidylinositol (GPI) anchor is an essential process that occurs in all eukaryotic cells. The role of this anchor is to direct newly synthesized proteins to the cell membrane. In fungi, some GPI-anchored proteins (GPI-APs) are also associated with the cell wall. Precursors of proteins that are modified by a GPI anchor share conserved features, including an amino-terminal signal sequence for localization to the endoplasmic reticulum (ER) and a carboxyl-terminal hydrophobic domain for attachment to the ER membrane [1]. Proteins are cleaved near the so-called omega site (Ω), several amino acids before this C-terminal, before ligation to a pre-assembled GPI anchor. These characteristics have been utilized to direct in silico searches of annotated genomes; recently, 86 GPI-APs were identified in the genome of the pathogenic fungus *Aspergillus fumigatus* [2].

In *A. fumigatus*, GPI-APs play a major enzymatic role in cell wall biosynthesis [2,3]. For example, the *GEL* family of β-(1,3)-glucanosyl-transferases (containing seven members) is responsible for the elongation and branching of β-(1,3)-glucans, an activity essential for fungal growth [4,5,6,7]. The *DFG* family with seven members is involved in the covalent binding of galactomannan (GM) to the β-(1,3)-glucan-chitin core of the cell wall [8]. Moreover, members of GH16 glucanase families are required to establish normal conidial structures [9], and Bgt2 is a predominant protein in the cell wall encoding β-(1,3)-glucan branching activity [10].

Comprehensive analyses of GPI-anchored proteins were undertaken in *Candida albicans* and *Saccharomyces cerevisiae* [11,12]. These studies have shown that these proteins are involved in fungal cell wall integrity, viability, virulence, chemical, and environmental stresses [13]. To date, similar studies have not been conducted in filamentous fungi. However, the investigation of a morphogenetic role unrelated to cell wall functioning has now become possible due to the availability of a mutant library based on the deletion of all genes coding GPI-APs. Three groups of GPI-APs have been recognized in *A. fumigatus* [2]. The first group of proteins is highly conserved in all fungi (yeast as well as filamentous fungi), which are essential in cell wall morphogenesis, most of them belong to multigenic families of proteins (e.g., *GEL*, *DFG*, *CRH* families). The second group of proteins is present only in filamentous fungi, which are mostly involved in the processes of biofilm formation, adhesion, and virulence (CFEM, MP proteins) [14,15]. Finally, the third group of proteins is only present in the *Aspergillus* species or the related species of mold (ex *CSPA*) [16,17]. These proteins seem to be required for the normal formation of conidia, but how they carry out this function remains unclear.

During the analysis of the library of GPI-anchored mutants, we identified and characterized a novel GPI-AP that we call SwgA (for SWollen and Germinated conidia), which is involved in germination, growth, and morphogenesis. Moreover, we show that *swgA* is involved in nitrogen metabolism under the regulatory control of the transcription factor AreA.

## 2. Materials and Methods

### 2.1. Strains and Growth Conditions

The *A. fumigatus* parental strain KU80pyrG+ [18], an *A. fumigatus* strain derived from the well-characterized clinical isolate A1163 that lacks non-homologous end joining (*Δku80*), was used as a host for transformation experiments and maintained on 2% malt agar slants at room temperature. For DNA extraction, cultures were grown in Sabouraud liquid medium (2% glucose + 1% mycopeptone). Conidia were collected from agar slants/plates after seven days of growth at room temperature using 0.05% Tween 20 aqueous solution. The minimal medium (MM) used contained 5 mM of ammonium tartrate (NH_4+_) and 1% glucose as the nitrogen and carbon source, respectively (MM-NH_4+_). To select the transformant, 150 µg/mL of hygromycin (Sigma, Saint Quentin Fallavier, France) or chlorimuron (100 µg/mL) was added to the medium.

### 2.2. Protein Sequence Analysis

Protein sequences and AFUA numbers were retrieved from the CADRE Genome Browser (http://fungi.ensembl.org/Aspergillus_fumigatus/Info/Index, accessed on 15 February 2019). Protein sequences were analyzed with SignalP 3.0 (http://www.cbs.dtu.dk/services/SignalP/, accessed on 15 February 2019), the big-PI fungal predictor (http://mendel.imp.ac.at/gpi/fungi_server.html, accessed on 15 February 2019), PredGPI (http://gpcr.biocomp.unibo.it/predgpi/pred.htm, accessed on 15 February 2019), and the TMHMM Server v. 2.0 (http://www.cbs.dtu.dk/services/TMHMM/, accessed on 15 February 2019). Blast analyses were performed using (https://www.yeastgenome.org/blast-fungal, accessed on 20 March 2019) and (https://blast.ncbi.nlm.nih.gov/Blast.cgi, accessed on 20 March 2019). Protein sequences were aligned using ClustalX 2.0.12.

### 2.3. Construction of the ΔswgA Mutant, and the Revertant Strain ΔswgA::swgA

The construction of the *ΔswgA* mutant was carried out as described by Zhao et al. [19] and verification of the mutant was performed by the Southern blot analysis by Samalova et al. [2]. Complementation of the *ΔswgA* mutant was carried out by the reintroduction of the parental copy of the gene flanks and the chlorimuron-ethyl resistance β-recombinase (Chlori^R^-β-rec), Valsecchi, et al. [20] at a 3′ flanking region as described in Appendix A (the primers are listed in Appendix A). *A. fumigatus* conidia were transformed by the electroporation method described by Sanchez and Aguirre [21] with subsequent modifications. Conidia were washed with water and 10^9^ conidia were inoculated in 125 mL of YG medium (0.5% yeast extract, 2% glucose), and incubated at 37 °C at 300 rpm for 4 h to obtain swollen conidia. Those conidia were recovered by centrifugation and then inoculated in 12.5 mL of YG medium (1% yeast extract, 1% glucose, 20 mM HEPES, pH 8.0) and incubated for 1 h at 30 °C at 100 rpm. Conidia were centrifuged and resuspended in 1 mL of cold 1 M sorbitol; 10 µg of DNA were added to 50 µL of conidial suspension, incubated for 15 min on ice, and transferred to 0.1 cm electroporation tanks. Electroporation was performed using the Bio-Rad gene pulser (Gene Pulser Xcell, Bio-Rad, Hercules, CA, USA) with the following parameters: voltage, 1 kV; capacitance, 25 microfarads; and resistance, 400 Ω. After transformation, cold YG medium was added to the conidia and incubated on ice for 15 min and then incubated at 30 °C at 100 rpm for 1 h and 30 min. Conidia were plated on minimal media and incubated at 20 °C overnight. Hygromycin (150 µg/mL) was added to the plates before incubation at 37 °C for two days. After cultivation of the revertant on a minimal medium containing 2% xylose, the excision of the chlorimuron-ethyl resistance cassette was conducted. This strain was used in this study.

### 2.4. Manipulation of DNA, RNA, Southern Blotting

Genomic DNA was extracted as previously described by Girardin et al. [22]. For the Southern blot analysis, 10 µg of digested genomic DNA were subjected to size fractionation on 0.7% agarose and blotted onto a positively charged nylon membrane (Hybond-*N*+; Amersham, GE Healthcare, Chicago, IL, USA). For DNA extraction, mycelium was grown for 16 h at 37 °C in Sabouraud liquid medium. For RNA extraction, the mycelium of each strain was grown for 27 or 30 h in a liquid medium and then disrupted with 0.5 mm of glass beads in 500 µL, and RNA was then isolated using the phenol–chloroform method. DNase treatment was performed after elution of the RNA from the column with Turbo DNA-free DNase (Ambion^®^, Foster City, CA, USA). The quality of RNA was controlled on Bioanalyzer RNA 6000 nano Chips (Agilent Technologie, Santa Clara, CA, USA). One microgram of total RNA was reverse-transcribed using a reverse transcriptase Bio-Rad kit (Iscript cDNA synthesis kit, Bio-Rad, Hercules, CA, USA) following the instructions of the manufacturer. The gene expression levels of *swgA* and *areA* (AFUA_6G01970) were quantified by PCR using the same amount of cDNA obtained in the different liquid media MM-NH_4+_, RPMI-1640+Gln (containing L-glutamine), RPMI-1640 (RPMI without L-glutamine) and MM, without ammonium tartrate (NH_4_^+^) and supplemented with 10 mM L-glutamine)(MM+Gln10) with the following program: 1 min at 98 °C, 30 cycles of 10 s at 98 °C, 20 s at 65 °C and 40 s at 72 °C; and a final extension of 10 min at 72 °C.

### 2.5. Conidiation, Conidial Morphology, and Germination

The conidiation rates were estimated following inoculation of 10^5^ conidia onto 3 glass-test tubes (20 mL) containing 2% malt agar (10 mL/tube). After 1 week of growth at 37 °C, conidia were recovered from malt agar slants using a 0.05% Tween 20 aqueous solution. Conidial suspensions were filtered on a 40 µm pore-sized sterile cell strainer (Fisher Scientific, Illkirch, France) and counted using a Luna dual-fluorescence cell counter (Mokascience SARL, La Madelaine, France). After washing with 0.05% Tween 20 water, the permeability of the conidia with respect to FITC was investigated by incubating 200 µL of an aqueous suspension of conidia (2 × 10^7^ conidia/mL) with 30 µL of FITC solution (0.1 mg/mL in 0.1 M Na_2_CO_3_, pH 9) overnight at 4 °C in darkness. The conidia were washed three times with 0.05% Tween 20 solution before being subjected to observation under a fluorescence microscope (Evos FL Life Technologies; excitation wavelength [λex], 470/22 nm; emission wavelength [λem], 510/44 nm). For Calcofluor White (CFW) staining, the conidial suspension was incubated with CFW solution (0.5 µg/mL) for 30 min at room temperature and observed with fluorescence microscopy (Evos FL Life Technologies; λex, 357/44 nm; λem, 447/60 nm). Conidial germination was followed microscopically every 30 min after 4 h of growth at 37 °C or 45 °C (inoculation of 10 μL of conidia at 10^6^ cells/mL on a glass slide with 2% agar malt medium). For the viability assays, conidia on agar slants were stored at 37 °C for one and two months, then conidial suspensions were recovered in 0.05% Tween 20 water and the viability of conidia was evaluated following dilutions and plating on 2% malt medium.

### 2.6. Growth and Susceptibility of the Mutant Strains to Congo Red and Calcofluor White

Mycelial growth was tested on different types of agar media (Sabouraud, Minimal Medium (MM-NH_4+_), malt 2%, and glucose 3%-yeast extract 1% (G3%/YE1%)) at 30 °C, 37 °C and 45 °C. For temperatures of 30 °C and 37 °C, 10^6^ conidia (5 µL) of each strain were spotted for up to 72 h. To investigate the colony-forming ability of the *ΔswgA* mutant at 45 °C, mycelial disks cut out from the perimeter of pre-grown colonies were spotted on malt and MM-NH_4+_ plates and incubated for 72 h at 45 °C. Growth was also tested in 30 mL of liquid of several types of media containing different nitrogen sources and amino acids: G3%YE1%, MM-NH_4+_, RPMI-1640, RPMI-1640+Gln, MM without NH_4+_, and supplemented by either L-glutamine 10 mM (MM+Gln10), glutamate 10 (MM+Glu10), 50 mM (MM+Glu50), or MM-NH_4+_ supplemented by glutamate 10 mM (MM-NH_4+_+Glu10) and incubated with shaking (150 rpm/min) at 37 °C for 24, 30, or 48 h. The mycelial dry weight was estimated every 24 h after drying at 80 °C until a constant weight was reached. Growth inhibition tests with Congo Red (CR) (50–240 µg/mL) and CFW (50–200 µg/mL) were performed on MM agar plates.

### 2.7. Hydrophobicity Measurements

One ml of water was added to a three-week-old malt glass-test tube culture and vortexed for 30 s. The water containing hydrophilic conidia was carefully recovered using a Pasteur pipet. One mL of Tween 20 water was then added to the tube, which was vortexed for 30 s, and the Tween 20 water containing hydrophobic conidia was recovered. The percentage of hydrophobic conidia was estimated from the ratio of conidia counted in the Tween 20 water solution vs. the total number of conidia as described by Valsecchi et al. [20].

### 2.8. Scanning Electron Microscopy (SEM) Analysis

The strains were cultured on 2% malt medium for 24 h at 37 °C and then for 6 days at room temperature. For each strain, 10^6^ conidia were inoculated in liquid glucose 3% yeast extract 1% medium for 5 h and 8 h at 37 °C, 150 rpm. Then dormant, swollen, and germinated conidia were washed in sterile water, centrifuged for 5 min at 2500× *g,* and then fixed with 2.5% glutaraldehyde and 2.5% paraformaldehyde in PHEM pH 7 buffer (120 mM Pipes, 4 mM MgSO_4_, 20 mM EDTA and 50 mM HEPES) for 1 h at room temperature and after overnight at 4 °C without agitation. For SEM, they were then sequentially fixed with 1% osmium tetroxide in PHEM pH 7 buffer for 1 h and then dehydrated in ethanol followed by critical-point drying. Samples were then sputter coated with Au-Pd and imaged with a field-emission scanning electron microscope (JSM-6700F, Joel^®^).

### 2.9. Adhesion Assays and Biofilm Quantification

The ability of hyphae to adhere to polystyrene was tested by incubating 10 µL of conidia (10^6^/mL) for 24 h in 24-well polystyrene plates (TPP, Thermo Fisher, Illkirch, France) in 1 mL MM-NH_4+_ supplemented with 0.01% Tween 20 at 37 °C. The plates were then washed three times with PBS 0.01% Tween 20 and the remaining adherent mycelium was quantified by adding 130 µL of crystal violet 0.01% for 20 min at room temperature, washed 3 times with water, and de-stained with 130 µL of 30% acetic acid. Growth was estimated by measuring OD (Optic Density) at 600 nm. Mycelia under aerial static biofilm or planktonic conditions were obtained as previously described [23]. Briefly, 10^6^/mL conidia were inoculated in 20 mL liquid, or agar MM-NH_4+_ covered by cellophane (DryEase cellophane, Invitrogen, Carlsbad, CA, USA) at 30 °C for 24 h, and the mycelium dry weight was quantified.

### 2.10. Anti-SwgA Antibodies

Rabbits were immunized against two different peptides SWLKRDTLKPFQEG-Cys and Cys-STRTRATDTDDVWE designed based on sequence data (ProteoGenix). Coupling of the peptide through cysteine to m-maleimidobenzoyl-N-hydroxy-succinimide ester, immunization of the animal, and titer determination of the antiserum was performed by ProteoGenix SAS (Schiltigheim, France). Immunopurification of the specific antipeptide antibodies was done after coupling the peptide to epoxy-activated Sepharose (Amersham Pharmacia Biotech, Orsay, France) following the instructions of the manufacturer. Immunolabeling of blots was done using the ECL Western blotting detection procedure of Amersham Pharmacia Biotech.

### 2.11. Immunodetection of SwgA in Cell Wall or Membrane

After 24 h of growth at 37 °C in G3%YE1% liquid medium, the mycelium of the parental strain, the *ΔswgA* mutant, and the revertant strain were disrupted using 1 mm beads for two minutes at 4 °C using a Fast-Prep cell breaker (MP Biomedical, Illkirch-Graffenstaden, France) in Tris-HCl50 mM buffer pH 7.5, 1 mM EDTA, 0.3 M sucrose, 1% glucose, 2 mM DTT including antiprotease (Complete Protease inhibitor cocktail, Roche, Bale, Swizterland). Cell walls were collected by centrifugation for 10 min at 4000× *g* and then washed three times with water. The cell lysate supernatants were centrifuged for 1 h at 36,000× *g* at 4 °C to pellet total cellular membranes which were then suspended in the extraction buffer and stored at −20 °C. Proteins from the cell wall and membrane fractions were extracted by SDS-mercaptoethanol buffer (Tris-HCl 62 mM pH 6.8 containing 2% SDS and 5% β-mercaptoethanol) and were analyzed by SDS-PAGE on a 12% polyacrylamide gel, under reducing conditions (Laemmli, Bio-Rad Mini-Protean Tetra Cell instruction manual). SwgA was detected by Western blot using anti-SwgA serum (1:1000 dilution) and secondary anti-rabbit IgG conjugated to peroxidase (1:2000 dilution; Sigma, Saint Quentin Fallavier, France). The detection was performed with the ECL chemiluminescence method of Amersham (GE Healthcare Life Sciences, Vélizy-Villacoublay, France).

### 2.12. Immunofluorescence of Cell Wall Polysaccharides

For each strain, 10^6^ conidia were inoculated in liquid G3%YE1% medium for 4 h (swollen conidia) at 37 °C, 150 rpm. After 5 min centrifugation at 4000 rpm, dormant and swollen conidia were washed in 1× PBS-0.05% Tween 20 (PBS-Tw). Briefly, conidia were fixed with p-formaldehyde 2.5% (pFA) overnight at 4 °C and washed with phosphate-buffered saline (PBS) pH 7 containing 0.1 M NH_4_Cl and then with PBS-Tw. After washing, conidia were resuspended in PBS-Tw supplemented with 5% Goat serum (PBS-Tw-G) for 30 min before adding the primary antibodies or receptors described below. After incubation for 1 h, the conidia were washed and incubated in PBS-Tw-G with the secondary antibodies conjugated to a fluorochrome. After washing in PBS-Tw-G, the samples were observed under a wide-field epifluorescence microscope. The different antibodies tested were the following: for SwgA detection, samples were immunolabeled with the polyclonal antibodies anti-SwgA (dilution 1:500) and a secondary anti-rabbit IgG conjugated to Alexa488 (anti-rabbit IgG-A488; Sigma). For chitin and glycoproteins detection, chitin and glycoproteins were labeled with wheat germ agglutinin coupled with tetramethylrhodamine (WGA-TRITC) and concanavalin A labeled with fluorescein isothiocyanate (ConA-FITC), respectively, (Sigma) at 0.1 mg/mL. The α-(1,3)-glucan was detected by immunolabeling with the anti-α-(1,3)-glucan IgA monoclonal antibody B-4.1 [kind gift of J. Kearney, University of Alabama, Birmingham, USA or with the MOPC IgM [Sigma, 8 μg/mL [24], and revealed by the anti-mouse IgG-A488. For the Anti-SwgA, the primary antibody was diluted 1/50, and the secondary FITC rabbit 1/200. Fluorescent images were collected using an EVOS FL epifluorescence microscope (Life Technology^®^) with a 1.3-megapixel charge-coupled-device (CCD) camera (Sony ICX455 monochrome).

### 2.13. Carbohydrate Analysis of the Cell Wall Fractions and Supernatant

The amount of 5 × 10^7^ conidia of the parental strain, *∆swgA* mutant, and the revertant strain were incubated for 24 h in 40 mL MM-NH_4+_ at 37 °C, 150 rpm. The mycelium was then collected by filtration under vacuum and washed with water and the culture supernatant was precipitated with 2.5 volume of ethanol overnight at 4 °C. The cell wall was prepared as described by Millet et al. [25]. Total neutral sugar content was estimated by the phenol sulfuric method [26] and glucosamines and galactosamines were quantified by high-performance liquid chromatography (HPLC) after acid hydrolysis with 6 N HCl at 100 °C for 6 h [27].

### 2.14. In Vivo Experiments in a Murine Model of Invasive Aspergillosis

Parental, *∆swgA* mutant, and *∆swgA::swgA A. fumigatus* strains were cultured in 2% malt agar in glass-test tubes. For virulence assays, twenty-four-week-old OF1 female mice (each 18–20 g, Charles River Laboratory, (L’Arbresle, France)) were immunosuppressed with Kenacort^®^ Retard (Acetonide de Triamcinolone; Bristol-Myers Squibb) injected subcutaneously on day −1 (40 mg Kenacort per kg of the mouse). Before conidial inhalation (day 0) mice were anesthetized with an intraperitoneal injection of 0.2 mL of a solution containing 10 mg/mL ketamine (Imalgène^®^ 1000, Lyon, France), 1 mg/mL xylazine (Rompun^®^, Sigma, Saint Quentin Fallavier, France) per mouse. By means of an automatic pipetting device, each mouse was inoculated intranasally with 2 × 10^6^ conidia (20 µL per mouse of 0.05% Tween 20 conidial suspension at 10^8^ conidia per mL). Non-infected control immunosuppressed mice only received 20 µL of PBS plus 0.05% Tween 20. Daily monitoring of the body weight was done and a loss greater than 25% of the initial body weight was considered as a limit of pain not to be exceeded. Any animal that passed the critical weight loss would be immediately euthanized. The simple survival analysis was done by the Kaplan–Meier test. The statistical significance of the comparative survival values was calculated using the Mantel–Cox and Gehan–Breslow–Wilcoxon log-rank analyses, and the Prism statistical analysis package.

### 2.15. Statistical Analysis

Statistical data analysis was performed with GraphPad Prism. Unpaired, two-tailed Student’s t tests or one-way analyses of variance (ANOVA) were used to determine statistically significant differences in growth-related, conidiation, or germination phenotypes. Differences between the groups were considered significant at a *p*-value of ≤0.05. Throughout the article, significance is denoted as follows: * *p* < 0.0332; ** *p* < 0.00221; *** *p* < 0.0002; **** *p* < 0.0001, ns, nonsignificant.

## 3. Results

### 3.1. A Unique GPI-Anchored Protein Common to Aspergillus Section Clavati

Our preliminary screen of the library of putative GPI-anchored proteins of unknown function in *A. fumigatus* showed that only one of them encoded by the gene AFUA_8G01170 was involved in fungal growth [2]. Its morphogenetic function was confirmed after the construction of the complemented strain using the chlorimuron β-rec/six cassette [20] (Appendix A). AFUA_8G01170 (AFUB_084830) coded for a putative GPI-anchored protein of 281 amino acid residues including a signal peptide of 23 amino acids, and a 27 amino acid hydrophobic C-terminus with the amino acid trimer SGA providing the Ω cleavage site (http://gpcr.biocomp.unibo.it/predgpi/pred.htm, accessed on 15 February 2019). Blast analysis revealed that orthologs of this protein were restricted to *Aspergillus* section *Clavati*. Alignment of the orthologues showed that the N-terminal 120 amino acids are highly conserved (61 to 90% identity) and rich in cysteine residues (17 conserved cysteine residues) (Appendix A). The characteristics of the protein encoded by AFUA_8G01170 are given in Appendix A. Although no significant homology was found with any known proteins, it was classified as a hydrophobin-like protein because of the presence of distinctive patterns of conserved cysteines and hydrophobic residues [28]. Analysis of the protein using the fungal adhesins and adhesin-like protein prediction tool (http://bioinfo.icgeb.res.in/faap/query.html, accessed on 15 February 2019) indicated that it was likely to be an adhesin [29].

Analysis of previously published RNA-seq data designed to assess transcription during conidial germination showed that AFUA_8G01170 is not expressed in dormant conidia, but its expression level was increased during the swelling and germination process (from 4 h to 8 h of growth) [9].

For this reason, the gene was named *swgA* for Swollen and Germinated conidia. The use of an anti-SwgA antibody has confirmed the gene expression during germination, SwgA was detected by immunofluorescence as a dot at the cell surface (Figure 1A). Western blots using the anti-SwgA on protein extracts from membrane and cell wall fractions of the wild type, revertant strain, and *ΔswgA* mutant conidia, as well as hyphae, confirmed that SwgA was mainly located in the membrane fraction. The protein migrated with a molecular marker of 37 kDa. The expected size of this protein was 31 kDa, suggesting that this protein could be glycosylated (Figure 1B).

### 3.2. SwgA Is Not Involved in Cell Wall Synthesis

No difference in hydrophobicity was observed for the *ΔswgA* mutant compared to the parental or the revertant strain (data not shown). CFW and FITC labeling can be used as a proxy to check the permeability of cell wall mutants [30] and revealed that the permeability of *ΔswgA* dormant and swollen conidia was similar to the parental and revertant strains (Appendix A). Since the role of GPI-anchored proteins has been often associated with cell wall biosynthesis [2], the cell wall of parental and *ΔswgA* mutant strains was analyzed. However, no discernable differences in hexose, N-acetylglucosamine, or N-acetylgalactosamine levels were observed (Table 1). Immunolabeling of swollen conidia of the parental and *ΔswgA* mutant was performed with antibodies specifically recognizing the polysaccharides of the cell wall and showed no differences in the localization of α-(1-3)-glucan and chitin in parental and *ΔswgA* mutant strains (Appendix A). Moreover, no significant modifications of the outside layer of the cell wall were observed in SEM (Appendix A). Altogether, these results indicated that the SwgA protein is not involved in cell wall organization.

### 3.3. The Growth Phenotype of the ∆swgA Mutant Is Temperature Sensitive

Important morphogenetic alterations occurred in the *∆swgA* mutant. First, a decrease in conidiation was observed for the mutant (35% ± 1) compared to the parental and the revertant strains (Figure 2A). Germination was delayed in the mutant: after 8 h of growth, 13% ± 3 of conidia had germinated in the mutant and 23% ± 1 in the parental strain, but all conidia were able to germinate (100% after 13 h of growth for both mutant and parental strains) (Figure 2B). This delay increased with temperature. At 45 °C, after 10 h of growth, only 9% ± 1 of mutant conidia germinated compared to 84% ± 2 for the parental or the revertant strain (Figure 2C).

The growth kinetics of the parental, the *ΔswgA* mutant, and the revertant strains were investigated on MM-NH_4+_ and malt agar media. After 48 h at 37 °C, a two-fold reduction of growth was observed for the mutant independently of the agar medium used (Figure 3A). The growth defect was more severe at 45 °C (>three times reduction of growth) (Figure 3B), confirming the temperature-sensitive phenotype observed for the mutant during conidiation and germination. Moreover, we observed a strong sensitivity to CR and CFW (Figure 3C). Considering that we did not observe any difference in permeability or cell wall composition (see Appendix A and Table 1) for the *ΔswgA* mutant, this strong sensitivity suggested cell wall structural differences with the parental strain.

A significant reduction of growth of the mutant compared to the parental strain was also observed in liquid glucose 3%/Yeast extract 1% medium, and in MM-NH_4+_ medium, at 37 °C (27% reduction after 48 h of growth for both media), however, surprisingly no reduction of growth was observed in RPMI-1640+Gln liquid medium (Figure 4). At 45 °C after 24 h, the growth defect was more pronounced, except for RPMI-1640+Gln medium, where no reduction of growth was observed (Appendix A). All these results suggested that the growth defect of the *ΔswgA* mutant was related to the nutrient composition of the medium. The ability of hyphae to adhere to polystyrene was also tested after 24 h of growth in liquid MM-NH_4+_ medium at 37 °C, the adherent mycelium was quantified by Crystal violet and the OD were observed at 600 nm. A growth decrease ratio of three times was observed for the *ΔswgA* mutant compared to the parental strain similar to the difference observed when the growth kinetics of both strains were analyzed indicating that the *ΔswgA* mutant did not show differences in adhesion (Figure 4 and Appendix A). Accordingly, no difference in biofilm formation was observed (data not shown).

### 3.4. swgA Is Involved in Nitrogen Metabolism and Is Regulated by the Transcription Factor AreA (AFUA_6G01970)

To investigate the putative function of SwgA in nutritional metabolism, the growth of the mutants was compared in several different liquid media at 37 °C after 27 h or 30 h of growth with different nitrogen sources: MM(without NH_4_^+^)(MM) and supplemented by either L-glutamine 10 mM (MM+Gln10), or glutamate 10 ((MM+Glu10) or 50 mM (MM+Glu50), also MM-NH_4+_ supplemented by glutamate 10 mM (MM-NH_4+_+Glu10), and RPMI-1640 without L-glutamine (RPMI-1640). Depending on the nitrogen source, differences in the growth of the *ΔswgA* mutant in comparison to the parental strain were observed (Figure 5A). A decrease of 35 to 50% of growth was observed for the *ΔswgA* mutant in comparison to the parental strain when MM-NH_4+_ or RPMI-1640 were used. In contrast, when MM+ Gln10 or RPMI-1640+Gln were used, no difference in growth was observed for the *ΔswgA* mutant (Figure 5A). In MM+Glu10 or MM+Glu50, the growth of *ΔswgA* was significantly decreased. In contrast, in MM-NH_4+_+Glu10, no growth defect was observed indicating that the phenotype was not due to a defect of glutamine synthase activity (Appendix A). The *∆areA* mutant was also included as a control [31] as AreA is a major regulator of nitrogen metabolism. Similar results were observed for the *∆areA* mutant indicating that the *ΔswgA* and *∆areA* mutants behave similarly to the wild type in a medium containing L-glutamine as the nitrogen source. The expressions of both genes were investigated in all of these media in the parental strain, and the *∆areA* mutant. In the parental strain, *swgA* and *areA* were not expressed in a medium containing L-glutamine (RPMI-1640+Gln and MM+Gln10 (Figure 5B). In contrast, these two genes were expressed in a medium without L-glutamine (MM-NH_4+,_ RPMI-1640). No *swgA* gene expression was detected in the *∆areA* mutant (data not shown) independently of the medium used, indicating that the transcription factor AreA regulated *swgA* expression. Moreover, a consensus sequence CGATA is present in 5′ position 313 nucleotides upstream of the start codon of *swgA* supporting the fact that *swgA* is one of the many genes regulated by the transcription factor AreA.

All of these results showed that *swgA* is involved in nitrogen metabolism and is regulated by the transcription factor AreA.

### 3.5. Virulence of the ΔswgA Mutant Is Affected

The virulence of the *ΔswgA* mutant was tested in a mouse model of invasive aspergillosis. The parental and the revertant strains showed the same survival curve. However, statistical analysis showed that the virulence of the *ΔswgA* mutant was significantly reduced (*p*-value < 0.5 (Figure 6).

## 4. Discussion

Our preliminary analysis of the *A. fumigatus* mutant library of all the GPI-AP encoding (identified via an in silico analysis) has shown that three mutants have significant reduction in the growth on malt and MM media at 37 °C [2]. Our screening was validated by the fact that *GEL2* and *DFG3* mutants of the library [5,8] had reduced growth. Only one mutant with a drastic reduction of growth, corresponding to the gene AFUA_8G01170 had not been investigated previously.

In this study, we characterized a new GPI-anchored protein that we named SwgA. Although no significant homology has been found with any known proteins, it has been classified as a hydrophobin-like protein because of the presence of distinctive patterns of conserved cysteines and hydrophobic residues as proposed by Littlejohn et al. [28]. Hydrophobins are low molecular weight proteins secreted by filamentous fungi and are divided into two classes (class I form functional amyloid fibers organized in layers with rodlet morphology, while class II hydrophobin layers show no defined morphology). In general, hydrophobins have low sequence conservation, and they are characterized by their hydrophobicity profiles and an idiosyncratic pattern of eight conserved cysteine residues that form four disulfate linkages. Seven hydrophobins have been characterized and studied, *RODA-RODG* in *A. fumigatus* [20,32]. It has been shown that only RodA is responsible for rodlet formation, sporulation conidial hydrophobicity, and immunological inertia of the conidia. Our data suggested that SwgA should not be considered a hydrophobin-like protein. For the *ΔswgA* mutant, we have not observed differences in rodlet formation or conidial hydrophobicity (data not shown).

Other proteins containing conserved cysteine residues, such as proteins with CFEM (common in fungal extracellular membrane) domains, are shown to be involved in host–pathogen interactions and virulence. In *C. albicans*, deletion of the three GPI-anchored CFEM-encoding genes in the genome (*RBT5*/*RBT51*/*CSA1*) resulted in an increased sensitivity to cell wall damaging agents and reduced ability to form a biofilm [33,34]. In the genome of *A. fumigatus*, three GPI-APs, CFEM (A-C), containing fungal-specific CFEM domains characterized by spaced cysteine residues are present [35]. Vaknin et al. [14] showed that these proteins did not play any role in cell wall morphogenesis or virulence. However, we did not observe any sequence similarity with CFEM proteins and SwgA.

The adhesion of *A. fumigatus* conidia to host tissues is an essential event leading to the persistence and infection of the host target lung. Several *A. fumigatus* molecules are shown to serve as host cell adhesion factors in the swollen conidia, germling, and hyphal states [36,37]. Takahashi-Nakaguchi et al. [38] identified 31 genes as putative candidates for conidial adhesion factors. The classification of SwgA as a hydrophobin-like protein, and its corresponding gene expression during hyphal morphogenesis, suggested a putative role in fungal adherence of this protein as predicted by Faapred’s prediction [29]. However, we demonstrated here that SwgA is not involved in adhesion and no difference in biofilm formation has been observed (data not shown).

In contrast, germination and growth of the *ΔswgA* mutant were strongly affected depending on the culture medium. We demonstrated that the growth of the *ΔswgA* mutant depends on the nitrogen composition of the medium. No differences in growth were observed between the *ΔswgA* mutant and the wild type in a liquid medium containing L-glutamine.

Nitrogen is an essential requirement for growth and fungi are able to use a wide variety of compounds as nitrogen sources. However, the use of different nitrogen compounds is selective, and readily assimilated nitrogen sources such as ammonium and glutamine are preferentially used [39]. In *Aspergillus*, the *areA* gene encodes the major nitrogen regulatory protein, which activates the transcription of many structural gene-encoding enzymes for nitrogen source catabolism under nitrogen-limiting conditions [40]. We showed that *swgA* expression is under the control of the transcription factor AreA. Indeed, we showed that *swgA* is not expressed in the *ΔareA* mutant whatever the medium used. Moreover, the AreA transcription factor, such as the GATA factor in fungi, share a common DNA binding motif that recognizes a core 5′-GATA-3′ sequence [41].

The growth of the *ΔareA* mutant in *A. nidulans*, *A. fumigatus*, and *A. oryzae*, in different nitrogen media were investigated [42,43,44]. The growth of these *ΔareA* mutants was identical to the wild type in presence of glutamine and the growth was affected by other nitrogen sources.

In our study, several nitrogen sources were tested and we showed that the *ΔswgA* mutant phenocopies the *ΔareA* mutant, i.e., in the presence of glutamine as the sole nitrogen source (RPMI-1640+Gln or MM+Gln), the growth of these mutants were not affected in contrast to the medium, which did not contain glutamine, such as the MM-NH_4+_ or RPMI-1640 medium. In *Aspergillus oryzae* and *Aspergillus nidulans*, the glutamine synthetase levels were not affected [42] by the *areA* deletion. The restoration of the growth phenotype of the *ΔswgA* mutant in the presence of NH_4_^+^ and glutamate suggested that the glutamine synthetase level, such as for *ΔareA* mutants, was also not affected in this *A. fumigatus* (Appendix A).

Consistent with the reduction in virulence observed for the *ΔswgA* mutant, loss of *areA* in *A. fumigatus* was shown to contribute to virulence but was not essential for virulence in a neutropenic murine model of pulmonary aspergillosis, with the deletion mutants presenting a delayed growth phenotype during infection [44]. In vivo data analysis showed that the gene expression level of *swgA* is 80 times higher than in vitro (Beau, Latgé, unpublished data) suggesting the lack of availability of preferred nitrogen source in the lung. This could explain the reduction of virulence of the *ΔswgA*. The mechanism involved in this nitrogen metabolism for SwgA is still unknown, but it was recently shown in *Pichia pastoris* that a GPI-anchored protein of unknown function, Gcw13, was able to regulate a permease GAP1 resulting in decreasing Gap1-dependent uptake of amino acids [45]. Moreover, in *C. albicans*, several permeases, such as Csy1p [46] and Gap2p [47], are shown to participate in glutamine assimilation. To date, several permeases are shown to be present in the *A. fumigatus* genome, but until now, not one has been characterized [48].

## 5. Conclusions

In this study, we showed that a new GPI-anchored protein called SwgA is involved in nitrogen metabolism; this protein is regulated by the transcription factor AreA. Our data showed that GPI-anchored proteins are not only involved in cell wall biosynthesis but may be key actors in the regulation of the general fungal metabolism.

## Figures and Tables

**Figure 1 jof-09-00256-f001:**
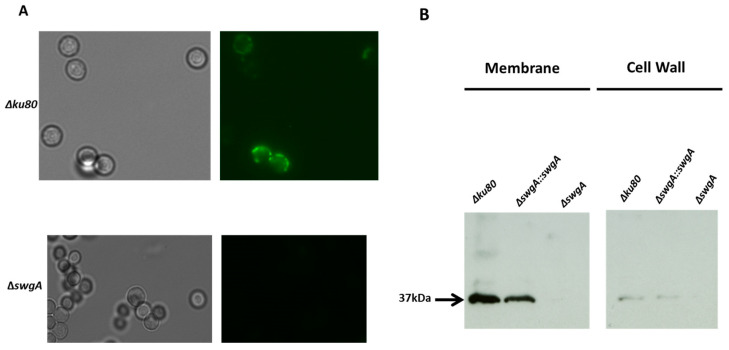
(**A**) SwgA localization by immunolabeling with the anti-SwgA antibodies (1/250) in swollen conidia (after 4 h of growth in glucose 3% yeast extract 1% liquid medium) of the parental and the *ΔswgA* mutant strains. (**B**) Western blot of the cell wall and membrane of the parental, complemented strain and *ΔswgA* mutant strains labeled with the anti-SwgA antibody (1/500). The membrane and cell wall were prepared after 24 h of growth in Glucose 3% Yeast Extract 1% medium.

**Figure 2 jof-09-00256-f002:**
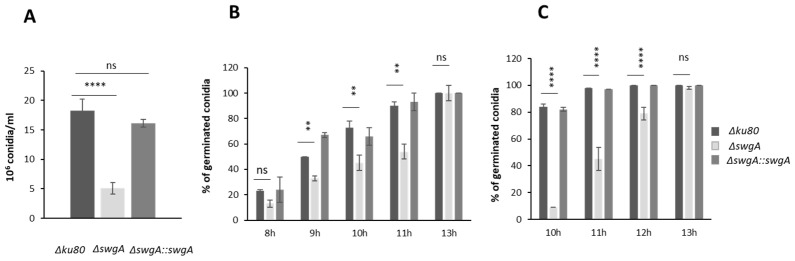
Conidiation and germination assay. (**A**) Conidiation of the parental, the revertant, and the *ΔswgA* mutant strains after one week at room temperature on malt medium. Kinetics of conidial germination: percentage (%) of germinated conidia of the parental strain, the revertant, and the *ΔswgA* mutant (**B**) at 37 °C on agar malt medium (**C**) at 45 °C on agar malt medium. The error bars represent the standard deviation from the mean values of three different experiments. *p* value are indicated: asterisks indicated that it is significant ** *p* < 0.00221; **** *p* < 0.0001 and ns, nonsignificant.

**Figure 3 jof-09-00256-f003:**
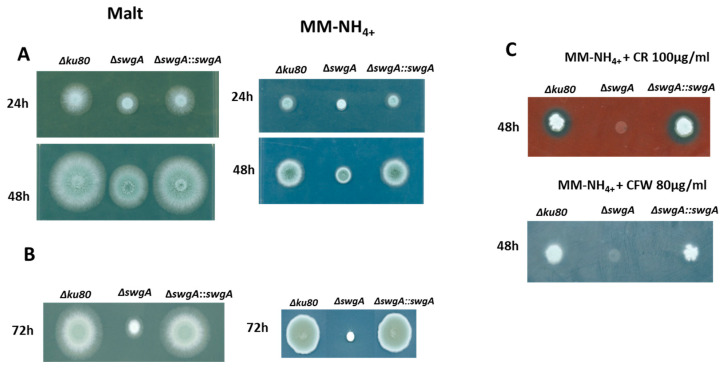
Growth of the parental, the revertant, and the *ΔswgA* mutant strains on malt agar and MM-NH_4+_ after 24 h, 48 h, and 72 h (**A**) at 37 °C (**B**) at 45 °C and (**C**) on agar MM-NH_4+_ medium after 48 h of growth at 37 °C with the addition of CR (100 µg/mL) or CFW (80 µg/mL).

**Figure 4 jof-09-00256-f004:**
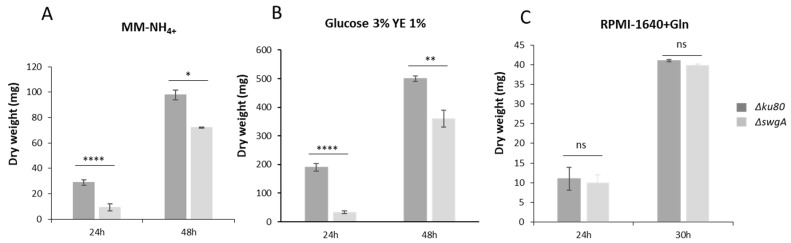
Growth kinetics of the parental and the *ΔswgA* mutant strains in (**A**) MM-NH_4+_, (**B**) glucose 3% yeast extract 1% (**C**) RPMI-1640+Gln liquid media after 24 h, 30 h, or 48 h of growth at 37 °C. *p* value are indicated: asterisks indicated that it is significant: * *p* < 0.0332; ** *p* < 0.00221; **** *p* < 0.0001 and ns, nonsignificant.

**Figure 5 jof-09-00256-f005:**
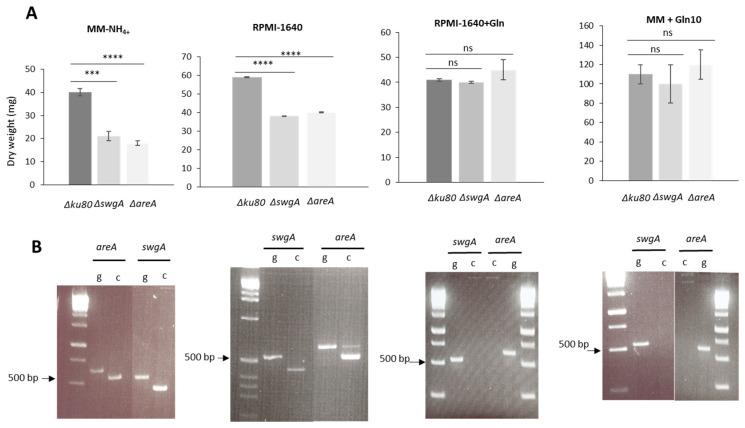
(**A**) Growth kinetics of the parental, *ΔswgA,* and *ΔareA* strains at 37 °C after 27 h in several different nitrogen sources: MM-NH_4+_, RPMI-1640, RPMI-1640+Gln, and MM+Gln10 and (**B**) gene expression levels of *areA* and *swgA* in the parental control strain in the different media shown above in panel A. g: genomic control, c: cDNA. *p* value are indicated: asterisks indicated that it is significant: *** *p* < 0.0002; **** *p* < 0.0001, ns, nonsignificant.

**Figure 6 jof-09-00256-f006:**
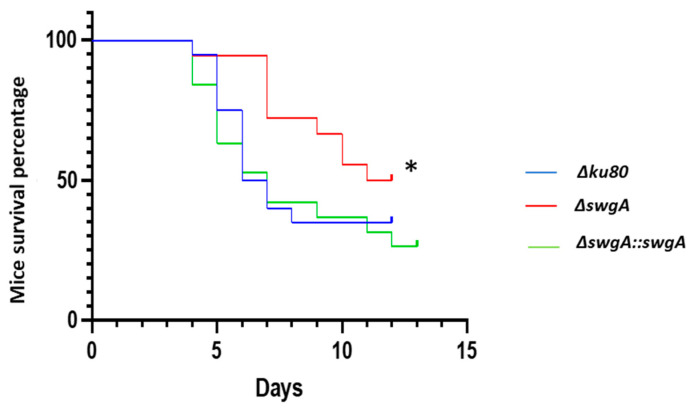
Mice survival percentage. Twenty-four-week-old OF1 female mice/strains were immunosuppressed with Kenacort. Before conidial inhalation (day 0), mice were anesthetized intraperitoneally with 10 mg/mL of ketamine and 1 mg/mL of xylazine per mouse. Inoculation was conducted intranasally with 2 × 10^6^ of conidia. Non-infected control immunosuppressed mice only received PBS (1X; ** p*-value Wilcoxon test < 0.5.

**Table 1 jof-09-00256-t001:** Percentage **(**%) of polysaccharide composition of the alkali-insoluble and the alkali-soluble fraction of the cell wall of the parental *Δku80* strain and the *ΔswgA* mutant. The cell wall alkali-insoluble fraction (AI) was purified from mycelium grown for 24 h in a liquid medium at 37 °C (expressed as a percentage of total hexoses plus hexosamines). Each value represents an average of data from three independent replicates. Hexose; GlcNAc, N-acetylglucosamine; GalNAc, N-acetylgalactosamine.

	Hexose	GlcNAc	GalNAc
*Δku80*-AI	66.85 ± 2.5	28.17 ± 1.9	4.98 ± 0.7
*ΔswgA*-AI	64.95 ± 5	30.31 ± 4.6	4.74 ± 0.54
*Δku80*-AS	42.63 ± 12.6	0	57.37 ± 12.7
*ΔswgA*-AS	27.54 ± 8.2	0	72.46 ± 8.2

## Data Availability

Not applicable.

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
