# Peer review of "The New GPI-Anchored Protein, SwgA, Is Involved in Nitrogen Metabolism in the Pathogenic Filamentous Fungus Aspergillus fumigatus"

_jof, 2023, doi:10.3390/jof9020256_

Round 1

Reviewer 1 Report

This is a very interesting manuscript reporting the biological functions of a novel GPI-anchored protein named SwgA, which is essential for germination, growth, and morphogenesis in the human opportunistic pathogen Aspergillus fumigatus.

However, the manuscript still has many writing errors (typos and grammar mistakes) that must be corrected. Please check carefully throughout the main manuscript and the supplementary data.

Some comments:

- In the abstract: two many "showed, shown"

- Line 99: the protocol for transformation by electroporation should be briefly described.

- Line 196: centrifugation for 10 min (with "for")

- Line 225: Check the citation format

- Line 228: Life Technology (not "Technlogy)

- Line 241: twenty-four weeks old (not "twenty 4")

- Line 247: Bayer HealthCare (not "Bayer Health- Care")

- Line 248: 2x106 (not "2.106")

- Lines 306-315: Check the position of Table 1. Currently, it is inserted in the middle of the paragraph.

- Consider changing the colors for the columns of graphs in Figure 3. This figure caption also needs a title.

- Check typos in the caption of Figure 4.

- Lines 470-473: consider rephrasing these sentences to make them more understandable.

- Conclusion: consider rewriting this section based on the obtained results of this work. The citations should not be used here. 

Author Response

Dear reviewer,

Thank you for your comments and corrections.

Please find enclosed our point by point response in attachment

Best regards

Reviewer 2 Report

The authors investigated the role of a so far unknown GPI-anchored protein, SwgA, presented in the cell membrane of A. fumigatus. The results are interesting and may be suitable for publication, but only after a major revision. 

Major concerns:

The statements about the role of SwgA in the abstract should be reconsidered. It is not accurate to write that SwgA is "essential for germination, growth and morphogenesis" (lane 26). It is an overstatement. According to the presented results, SwgA only plays role in germination, growth and morphogenesis, but there is no result that would support its essentiality in the above processes.

The data presented in Figure 1 was not obtained by the authors, but rather extracted from a public RNAseq database, therefore it should not be presented in such a way. The description of the phenomenon in the main text (lines 274-276) would be completely satisfactory. Hence, I suggest to remove Figure 1.

The authors do not follow the Aspergillus nomenclature for genes and proteins. Gene and protein naming is not consistent through the entire text and on the Figures. Besides mixing up yeast and Aspergillus nomenclature, gene and protein names are also interchangeably used in an inappropriate way, that should be carefully revised. Please, follow the accepted Aspergillus nomenclature when referring to genes and proteins according to the corresponding website: https://www.aspergillus.org.uk/genomes/a-proposal-for-the-naming-of-genes-in-aspergillus-species/. Briefly - SwgA: protein, swgA (in italic style): wild-type gene, DswgA or swgAD (in italic style): deleted gene, DswgA::swgA or swgAD::swgA or DswgA::swgA+ or swgAD::swgA+ (in italic style): reconstituted deletion.

According to my suggestions regarding the gene/protein names and media names, please make all corrections not just in the main text, but also in the supplemental materials.

The authors executed a germination ability analysis by monitoring the number of germinated conidia in a microscope. According to the literature (e.g. PMID: 31022275 paragraph „Ascospore content of cleistothecia and viability of ascospores in veA+ background”) the germination of spores observed in a microscope does not necessarily correlate with the colony forming ability. The colony forming ability correlates with the viability and not the germination ability. However, the authors did not examine the colony forming ability of the conidiospores in this work, which would have resulted in a different conclusion.

The authors found a considerably huge delay in the germination of DswgA conidia compared to the control at 45 °C (Figure 4), but they did not investigate the colony forming ability of the DswgA conidia on this high temperature. Results of such experiment would be useful to accurately chose the inoculation method for the growth tests. Testing various strains with significantly altered germination properties can be done by inoculating conidiospore-free mycelial disks (cut out from the perimeter of pregrown colonies) instead of inoculating conidiospores. Mycelial disk inoculation circumvent the adverse effect of altered germination properties on the outcome of the growth test. If there is indeed a difference in the colony forming ability between the control and DswgA conidia, I see no other option than conducting the growth test using the mycelial disk inoculation method.

Statistical analysis to assess significance is missing from Figure 3, 5 and 6.

I did not find any mentioning or discussion of the result presented in Figure 4c in the main text about the higher CR and CFW sensitivity of the DswgA strain compared to that of the swgA+ control. This is something that needs to be intrepreted to some extent in the manuscript as it may reflect structural differences in the cell wall composition or membrane integrity.

Did the authors performed paired Wilcoxon test for the analysis of results presented in Figure 7? I might be wrong, but a Cox proportional hazards regression seems to be a more appropriate analysis in a survival analysis.

The authors write in the Conclusion section  the followings: " In this study, we showed that a GPI anchored protein is involved in nitrogen uptake, this protein being regulated by the transcription factor AreA (lines 485-486)" and " The mechanism involved in this nitrogen uptake for SwgA is still unknown,..... (lines 488-489)".

1. The co-regulation of areA and swgA is supported by this work, however the regulation of swgA by areA is not verified. Regulation of swgA by areA is just indicated by the result of an in silico analysis showing the presence of an AreA binding site in the promoter region of swgA.

2. According to the literature, AreA-regulated genes fall under the repression by ammonium. How the authors discuss their own results that swgA seems to be well expressed in MM with ammonium as a sole nitrogen source (shown in Figure 6b)?

3. The involvement of SwgA in nitrogen uptake is a pure speculation. No experiments were carried out in this presented work to draw such a conclusion.

Due to the above critical comments the Conclusion section needs to be reconsidered.

Additional comments:

Several parts of the manuscripts need careful rewriting. See details below:

line 24: SwgA

line 27: swgA is controlled by the nitrogen regulator AreA.

lines 71, 93 and all over the text: SWGA denotes the wild type gene according to the yeast gene name nomenclature. According to the official Aspergillus nomenclature (https://www.aspergillus.org.uk/genomes/a-proposal-for-the-naming-of-genes-in-aspergillus-species/) the wild type gene is swgA. One may emphasize the wild type nature of a gene by placing a + symbol in superscript (swgA+). Therefore the reconstituted deletion strain should be DswgA::swgA or DswgA::swgA+.

line 103-119: How was the integrity of the RNA samples tested?

lines 114-115: „The gene expression levels of SWGA and AreA (AFUA_6G01970) were quantified by.....

The gene expression levels of swgA and areA (AFUA_6G01970) were quantified by.....

line 116: MM is not defined. Carbon and nitrogen sources should be provided. Was the N-source 5 mM ammonium-tartrate?

line 116: “RPMI-1640 (containing L-glutamine), RPMI-1640 (RPMI 116 without L-Glutamine)”

The formulation of the two media is not distinctive. For the sake of simplicity and straightforwardness, I suggest the formulation: RPMI-1640 and RPMI-1640+Gln (RPMI -1640 with L-Glutamine).

line 117: “MM, without NH4 and supplemented with 10 mM L-glutamine”

MM-Gln (MM with 10 mM L-glutamine as sole N-sorce).

Accordingly, the MM with 5 mM ammonium-tartrate as sole N-source would be abbreviated as MM-NH4+.

line 118: 1 min

line 121: Tube is too general for a description. Please define the exact tube used in the experiment (e.g. 2 ml microcentrifuge tube, 15/50 ml plastic centrifuge tube, 50 ml glass centrifuge tube).

lines 127, 248 and 406 : 2 ´ 106

lines 128, 198, 279, 310, 331, 332: 4 °C or 37 °C or 45 °C

line 143: 72 h

lines 145-147: “MM, RPMI-1640 (containing L-glutamine), RPMI-1640 (without L-glutamine), MM without ammonium tartrate and supplemented by either L-glutamine 10 mM, or glutamate 10 or 50 mM or MM with ammonium tartrate 5 mM supplemented by glutamate 10 mM”

According to my previous suggestions, please change the text to the following:

MM-NH4+, RPMI-1640, RPMI-1640+Gln, MM-Gln, MM-Glu-10 and MM-Glu-50 (minimal media with 10 mM and 50 mM glutamate as sole N sources, respectively), MM-NH4+-Glu (MM with 5 mM ammonium tartrate as N-source supplemented by 10 mM glutamate).

Using the suggested abbreviations for the used media through the entire text and on the figures would make the text easier to follow.

line 150: “Calcofluor White (CFW)”

CFW

line 151: “on MM agar plates.”

MM-NH4+ plates?

line 153: “Malt tube culture”

Define the tube accurately!

line 167: 1 h

line 175: delete with (washed with 3 times)

line 191: SwgA (not SwgAp)

line 195: 1%

line 203: SwgA

line 213: “” Goat 5 %”

Subject is missing: 5% Goat serum?

line 218: SwgA

line 226: AcSwgA stands for what?

line 240: DswgA::swgA

line 241: which kind of tubes?

line 241: twenty

line 245: intraperitoneal injection

line 246: ketamine

line 251: delete mice

line 252: …that passed the critical weight loss would be…

line 264: Aspergillus section Clavati

line 275: level was increased

line 278: swgA

line 282: swgA

line 284: SwgA

line 285: anti-SwgA

line 291: SwgA

Figure 2 between lines 289 and 290,Figure 3 between lines 327 and 328, Figure 4 between lines 339 and 340, Figure 7 between lines 401 and 402: DswgA::swgA+

line 296: SwgA

line 298: CFW

line 306: replace CW with cell wall

line 309: DswgA

lines 312-313: The authors say that statistical differences as ns (not significant), *p<0.01 and error bars are presented in Table 1. None of it are presented in the Table. These lines should be removed!

line 317: SwgA

line 344: "A significant reduction of growth of the mutant compared to the parental strain was also observed in liquid Glucose 3%/Yeast extract 1% medium, and in MM medium, ...."

The authors did not mention that they carried out statistical analysis to assess whether a difference observed between strains is significant or not. Statement of significant differences must be coupled by providing the P value.

line 347 and on Figure 5: RPMI-1640-Gln

line 362: swgA

line 364: SwgA

lines 377-378: MM-NH4+-Glu

lines 379-380: Rephrase the sentence! An areA deletion mutant is not a major regulator of nitrogen metabolism. The AreA is.

line 384: ....swgA and areA were not expressed.....

line 387: swgA

line 389: swgA

line 390: swgA

Figure 6a: Use the abbreviated names of media in the title of the graphs and correct the gene name of areA on the X axis (DareA instead of DAreA).

Figure 6b: Naming of the lanes is complicated. Simplify it.

line 394: sources

lines 394-395: MM-NH4+, RPMI-1640+Gln, RPMI-1640 and MM-Gln

line 396: ....of areA and swgA in the parental control strain in the different media shown above in panel A.

line 403: "Twenty 4 weeks old twenty OF1 female mice/strains"

Remove the second twenty.

line 427: SwgA

line 433: (RBT5/RBT51/CSA1)

(Yeast nomenclature for wild type gene names)

lines 439, 444,447: SwgA

line 460: swgA

line 462-464: The sentence would better fit in the Results section.

line 475: ammonium

Or if the authors insist on using the formula of the compound, then: NH4+.

lines 480-482: " In vivo data analysis showed that SWGA is 80 times higher than in vitro (Beau, Latgé, unpublished data) suggesting the lack of availability of preferred nitrogen source in the lung. "

Do the authors refer to the transcript of swgA or to the SwgA protein? It must be clarified. The used style here (SWGA) does not comply with either the gene or the protein nomenclature of an Aspergillus species.

Supplementary Figure S1: The Southern blot images and the explanatory drawings are not consistent.

1. NcoI digestion is marked above the filter only on the left panel, whereas it is missing on the right panel.

2. The sizes of the hybridizing DNA fragments are not formulated in English. Instead of 3,7 and 1,8 kb, the numbers should be written as 3.7 and 1.8 kb.

3. The RB region does not contain NcoI site in the swgA+ and DswgA locus in the left panel and also in the DswgA locus of the right panel. Please explain how it is possible that the same RB region contains NcoI site in the complemented DswgA locus!

4. The title of the top drawing in the right panel is missing.

Author Response

Dear reviewer,

Thank you for comments and corrections.

Please find enclosed our point by point response in attachement

Best regards

Round 2

Reviewer 2 Report

My concerns have been properly addressed. I support publication of this manuscript.

Please, find below a list of typos I found in the revised manuscript.

line 161: For temperature

line 161: 30 °C, 37 °C

line 162: 45 °C

line 164: 72 h, 45 °C

line 285:  delete * between p<0.0001 and comma

line 334: parental and

Table 1 between lines 341-342: Decimals must be notated with dots (for example, 66.85) instead of commas (66,85).

line 386: similar to

line 427: c: cDNA

line 495: oryzae

line 496: these – remove the italic format

line 511: the gene expression level of – remove the italic format

Legend of Table S3 in the Supplementary Material: Percentage

Author Response

Dear reviewer,

Please find the new version of the ms and sup data with the corrections.

Best regards